# Dental Fear and Anxiety of Chinese Preschool Children in a School-Based Outreach Service Using Silver Diamine Fluoride for Caries Control: A Cross-Sectional Study

**DOI:** 10.3390/ijerph20054467

**Published:** 2023-03-02

**Authors:** Ivy Guofang Sun, Hollis Haotian Chai, Edward Chin Man Lo, Chun Hung Chu, Duangporn Duangthip

**Affiliations:** Faculty of Dentistry, The University of Hong Kong, Hong Kong 999077, China

**Keywords:** caries, children, fluoride, fear, anxiety, silver diamine fluoride

## Abstract

Limited data are available regarding the patient-based outcomes of SDF therapy in the kindergarten setting. This study aims to evaluate the dental fear and anxiety (DFA) of preschool children after participating in a school-based outreach service using SDF to arrest early childhood caries (ECC). The study recruited 3- to 5-year-old children with untreated ECC. A trained dentist performed a dental examination and applied SDF therapy to the carious lesions. ECC experience was measured using the dmft index. Questionnaires for parents were used to collect the children’s demographic information and their dental treatment experiences. The children’s DFA before and immediately after SDF therapy was assessed using the self-reported Facial Image Scale (FIS) on a Likert scale of 1 (very happy) to 5 (very distressed). The association between the children’s DFA after SDF therapy and the potentially related factors, including demographic background, caries experience, and DFA before SDF therapy, were analysed using bivariate analysis. Three hundred and forty children (187 boys, 55%) joined this study. Their mean (SD) age and dmft scores were 4.8 (0.9) and 4.6 (3.6), respectively. Most of them (269/340, 79%) never had a dental visit. After SDF therapy, 86% (294/340) of the children exhibited no or low DFA (FIS ≤ 3), whereas 14% (46/340) reported high DFA (FIS > 3). No factor was associated with children’s DFA after SDF therapy (*p* > 0.05). This study showed most preschool children with ECC exhibited no or low DFA after SDF therapy in a school setting.

## 1. Introduction

Early childhood caries (ECC) is characterized by the presence of one or more decayed, missing or filled surfaces (dmf surfaces) in any primary teeth of a child aged five years old or younger [1]. It is one of the most prevalent diseases in young children. According to the global burden of disease, 532 million children have untreated caries in their primary teeth [2]. In Hong Kong, nearly half of preschool children (46%) have dental caries, whereas most of them (95%) remain untreated [3]. Children with untreated ECC may suffer from toothache, swelling and even facial cellulitis. In addition, poor oral health can affect children’s school performance [4] and quality of life [5]. Generally, the traditional treatment for dental caries is restorative treatment using restorative materials to fill the prepared cavities. However, the restorative approach alone is insufficient for caries treatment, especially in disadvantaged communities where the majority of children have untreated ECC. Furthermore, restorative treatment is a lengthy and complex process, making it challenging to gain young children’s cooperation [6].

With a further understanding of dental caries, there are currently two approaches for caries management; the surgical model involves restorative treatment, and the medical model primarily eliminates caries using preventive methods. According to the medical model, dental tissues afflicted by caries can heal if a non-cariogenic environment is maintained and remineralizing agents are used [7]. Following this model, non-invasive methods using silver diamine fluoride (SDF) have recently attracted more attention in community dental care [8].

SDF agent is a colourless alkaline solution containing silver and fluoride ions. Laboratories studies showed that silver ions promote the inhibition of bacterial growth and dentine collagen degradation [9], and fluoride has proven to be effective in remineralizing dental hard tissues by reacting with hydroxyapatite to form fluorohydroxyapatite on the tooth surface [10,11]. After the U.S. Food and Drug Administration cleared SDF to treat tooth sensitivity in 2014, SDF has been used off-label for caries arrest [12] and has become increasingly popular in recent years due to its minimal invasiveness. Several clinical trials confirmed that SDF at 38% can effectively arrest dental caries among children [13,14] and there was no major adverse effect associated with SDF therapy, except the black staining on the treated carious lesions [15].

Overall, SDF is a useful tool for managing ECC when used appropriately, but it should not be used as a one-size-fits-all solution. SDF may not be a permanent solution as the treatment is often required a regular annual reapplication and monitoring to continue its effectiveness [16]. Patients and their parents should be fully informed of the potential risks and benefits of using SDF before they decide to pursue this treatment. Although SDF therapy is efficient in arresting dental caries, treatment acceptability in young children is also critical.

Fear and anxiety are common emotions among children that can impede the successful completion of dental treatments. Studies have indicated that psychological factors, such as fear and anxiety, can affect the acceptability of dental treatment in children [17]. Good dental experiences during childhood can alleviate dental anxiety and help children develop oral health awareness, which will benefit them throughout their entire lives [18,19]. In reverse, an unpleasant dental experience could lead to DFA, and the effects of this anxiety could persist into adulthood, which can often lead to avoidance of dental treatment and subsequent deterioration of oral health [20,21].

The DFA prevalence among children varied between different studies depending on the study environment, therapy for oral disease and DFA measurement methods [22]. Most studies on DFA in children were assessed in a clinical setting after children received conventional restorative treatment or tooth extraction [23,24]. Few studies have reported DFA in children following SDF treatment. A survey reported DFA among children in a school-based setting using the Frankl Behaviour Rating Scale, a dentist-reported DFA outcome [25]. This observational rating scale reported by a dentist may have overestimated or underestimated children’s DFA when compared to child-reported DFA [26]. In addition, another SDF clinical trial evaluated the DFA using the child-reported assessment for DFA, the Facial Image Scale (FIS). Nevertheless, this trial was conducted in a clinical setting [27]. Thus, the results may not reflect the DFA of children using SDF therapy for caries control in a school-based environment. As is commonly known, SDF has drawbacks, including an unfavourable metallic and bitter taste, possibly affecting children’s acceptability and satisfaction with SDF therapy. Hence, it is necessary to investigate the children’s DFA after SDF treatment in a school-based setting.

Recently, SDF has been added to the World Health Organization’s (WHO) list of essential medicines, and the WHO recommends using SDF for caries control in the community [28]. Despite the proven efficacy of SDF therapy for caries control, limited data are available regarding the patient-based outcomes of SDF therapy in the kindergarten setting. Therefore, this study aimed to evaluate the DFA of preschool children with ECC after SDF therapy in a school-based outreach service and associated factors.

## 2. Materials and Methods

This cross-sectional observational study was conducted in Hong Kong from December 2019 to May 2021. This study’s reporting followed the STROBE statement’s guidelines [29].

### 2.1. Sample Size

The published clinical trial showed that the dental fear prevalence of children with dental caries was 17% in school-based settings [30]. The margin of error was set at 4%. With a statistical significance of 5% (α = 0.05), the required sample size was 339. According to the published research, the prevalence of untreated ECC among preschool children was 44% [3]. With an estimated response rate of 75%, at least 1028 children needed to be invited to participate in this study.

### 2.2. Population and Sample

This study recruited preschool children aged 3–5 years using stratified cluster sampling [3]. Because almost 95% of preschool children in Hong Kong attend kindergarten, the unit of sampling was a kindergarten. According to the number of preschool children in three main geographic areas in the Hong Kong, the ratio of invited schools in New Territories (NT), Kowloon (KL) and Hong Kong Island (HK) was 4:2:1.

Four kindergartens in NT, two in KL and one in HK were selected using a simple random sampling method using a list of computer-generated random numbers. After the kindergarten’s principal agreed to participate, a consent information letter was sent to the parents, containing the service’s purpose and procedures. The inclusion criteria were generally healthy children with untreated dental caries. Children who required special health care needs with severe cooperation or who were absent on the day were excluded.

### 2.3. Clinical Examination, SDF Treatment and DFA Assessment

A trained dentist performed clinical examinations in the kindergartens. In this study, a self-reported type of instrument for DFA assessment—the FIS—was chosen [26]. When the children entered the checking room, usually the music room or reading room of their kindergarten, they were asked to select one face from five that reflected their current mood. These five faces’ facial expressions ranged from very unhappy to very happy [31] (Table 1). The scores were recorded and later categorized for analysis: 0 = no anxiety (FIS = 1), 1 = low anxiety (FIS = 2 or 3) and 2 = high anxiety (FIS = 4 or 5) [32].

Then the children were asked to lie down on a table with their teacher’s help. Oral examination was performed using disposable dental mirrors (MirrorLite; Kudos Crow Limited, Hong Kong) connected to an illuminated intraoral handle and ball-ended WHO probes. Dental health status was recorded by using the decayed, missing and filled teeth (dmft) index according to the WHO criteria [33]. The use of dmft index could make comparison between populations as it was adopted in many studies worldwide.

The children with untreated dentine caries were to be treated with an 38% SDF agent (Saforide; Toyo Seiyaku Kasei Co., Ltd., Osaka, Japan). Before SDF treatment, the decayed tooth was isolated with cotton rolls, and then food debris and soft plaque were removed with cotton rolls. After using cotton roll or gauze to isolate the carious tooth, a dentist used a microbrush to dip SDF and then applied SDF to each carious lesion surface without removing the decayed tissue.

SDF was applied gently and made minimize the contact of SDF solution with gingiva or mucosa to avoid potential soft tissue irritation. The dentist then instructed the teachers not to allow children receiving SDF treatment to eat or drink for 30 min. Immediately after the treatment, the children were asked to complete the same FIS assessment again.

A research assistant recorded all clinical data on a paper sheet. To assess intra-reliability and reproducibility, approximately 10% of the participating children were randomly re-examined for caries status by the same dentist on the day of the dental visit.

### 2.4. Questionnaire Survey

The questionnaire was sent to the child’s parents before the outreach service to collect the information that have potential related association with children’s DFA. The questionnaire explored: (a) The demographic information of the child and parents, such as sex, age, birthplace, the child’s medical history, the parents’ education, and monthly household income level; and (b) The child’s and parents’ dental visit experience. Regarding to the child’s and parents’ dental visit experience, the parents were asked whether their child ever see a dentist and whether they ever visit a dentist.

The teachers distributed the questionnaire to the parents. The parents returned the signed consent form and the completed questionnaire before the day of the dental outreach service. An assistant checked the collected questionnaires and followed up on any missing or unclear answers via phone.

### 2.5. Data Entry and Statistical Analysis

Data input was performed using Excel software with a double check. Data were cleaned and then transferred to SPSS version 27.0 (IMB Corp., Chicago, IL, USA) for further analysis. Cohen’s kappa coefficient (κ) was used to calculate intra-examiner agreement on the caries assessment (dmft index). DFA-related potential factors were described. Bivariate analysis between the DFA level before and after SDF therapy was conducted using the McNemar test. The associations between other factors and children’s DFA after SDF therapy were analysed using the chi-square test. The significance level of all statistical tests was set at *p* = 0.05.

## 3. Results

A total of 1136 children aged 3- to 5-years old from 7 kindergartens were invited to participate in this study. Written informed consent was obtained from 871 children’s parents. Of these children, 529 children were excluded, due to not having untreated dental caries (n = 433), school absence on the day of dental visit (n = 83) or having special needs with severe cooperation (n = 13). Thus, 342 met the inclusion criteria, received SDF treatment and participated in the FIS assessment. Two parents did not return the questionnaires. Thus, 340 children were recruited for the study. A flowchart of the children’s recruitment is shown in Figure 1.

The mean age (SD) of the recruited children was 4.8 (0.9) years. The proportion of 3- to 5-year-old children was 22% (n = 74) for 3-year-olds, 31% (n = 106) for 4-year-olds, and 47% (n = 160) for 5-year-olds. Almost half of the children (55%, n = 187) were boys, and most (86%) were born in Hong Kong.

Their mean dmft (SD) was 4.6 (3.6), ranging from 1 to 19, and 139 (41%) of the children had a dmft score above 4 (dmft > 4). Most children (n = 269, 79%) never had a dental visit, while approximately one-quarter of the children’s parents (n = 89, 26%) never had a dental visit.

Before SDF therapy, most children (n = 297, 87%) had no or low DFA. In addition, most parents had tertiary education (mother and father, 80% and 78%, respectively). Table 2 shows the characteristic data of the children and parents. The Cohen’s kappa value for the caries assessment was 0.98.

According to the FIS assessment, 65% and 14% of children reported that they were very happy and happy after SDF therapy, and 7% were neutral toward the treatment, whereas 3% and 11% felt distressed and very distressed, respectively. The distribution of DFA before and after SDF treatment were showed in Figure 2.

For the bivariate analysis, children were classified into “no or low DFA” (FIS ≤ 3) and “high DFA” (FIS > 3). The result was that 86% (294/340) of children had no or low DFA (FIS ≤ 3), whereas 14% (46/340) reported high DFA (FIS > 3). The results of bivariate analysis between child’s DFA and potential factors are shown in Table 3. No significant difference was found in the relationship between children’s DFA after SDF therapy and potential factors including children’s information (sex, age, birthplace, caries experience, dental visit experience and DFA before SDF therapy) and parents’ information (dental visit experience, education level, and monthly income) (*p* > 0.05).

## 4. Discussion

In this study, a self-reported FIS instrument was used to reflect children’s DFA after SDF therapy. Results indicated that the prevalence of DFA (FIS > 3) in preschool children after SDF therapy in kindergarten was 14%, which was lower than the global average (22%) [34]. The following reasons may account for the low prevalence of DFA: The study was conducted in a familiar school setting with the presence of classmates and teachers. Similarly, a previous study reported that children who were treated in a familiar environment had lower DFA than those in a clinic setting [31]. In addition, due to the simplicity of SDF treatment, which does not require caries removal or local anaesthesia, most children were able to tolerate SDF treatment and show low DFA. This is consistent with the published study reporting that children who underwent simple procedures, such as prophylactic treatment, had lower DFA than children who received surgical treatment [35].

Interestingly, most of the children (86%) felt happy after receiving SDF therapy in this study, even though most of them had never visited a dentist. Possibly, using SDF for caries control as the first dental visit in kindergarten provides children with a positive dental experience and may help them avoid the DFA vicious cycle, in which patients with high DFA are more likely to postpone dental treatment, resulting in more severe dental problems [36]. Good early dental experiences may play a crucial role in determining DFA in adulthood. According to one review, positive dental treatment experiences prior to severe and complex treatments may reduce the likelihood of DFA [37]. SDF therapy, as a non-invasive treatment, could be a good starting point for children’s lifelong dental care.

Regarding the influence of previous dental experiences on DFA, several studies have shown that most adults or adolescents with DFA avoided dental treatment due to negative experiences during their early dental visits [17]. In the current study, the prior DFA of the study children was not statistically associated with their DFA after SDF treatment. This is in accordance with the results of the study in Brazil, which showed there were no significant changes in DFA before and after SDF treatment when it was performed in the dental clinic [27]. Note that, in our study, the majority (70%) of children with high DFA before SDF therapy changed to no or low DFA after SDF therapy. Some children exhibited high DFA even before the dental examination because the young children could perceive the dental environment as an overwhelming, anxiety-provoking sensory experience [38]. However, due to the simplicity of SDF treatment with no caries removal, no drilling, and no local anaesthesia administered, most of these children who exhibited high DFA before SDF treatment accepted the treatment and felt happy or very happy at the end of treatment. From this point of view, using SDF in outreach dental services for caries control may not only alleviate dental pain and infection but may also assist children who are extremely anxious about dental treatment in alleviating their DFA in the future.

According to other studies, younger students had a higher DFA than older students did [32,39]. Contrary to the current study’s results, the relationship between age and DFA was not observed in our study. Surprisingly, our study showed that 3-year-old children had a slightly lower DFA than 5-year-old children did (14% vs. 17%). This might be because the lower number of decayed teeth of the younger children required a shorter treatment time, which resulted in their post-treatment DFA being lower than that of the older children with more decayed teeth.

The present study had several strengths. Firstly, a stratified cluster sampling method was used to select kindergartens in different geographical areas representing the population of preschool children in Hong Kong. Therefore, this study reflected the status of DFA distribution among preschool children after SDF therapy in Hong Kong. In addition, this was the first study to focus on the DFA assessment using a child-reported scale (FIS) after SDF therapy in school-based settings. It can reflect the children’s DFA after SDF therapy for caries control in the school-based outreach dental service.

However, also note that there were some limitations in this study. This study was a cross-sectional study in which children’s DFA and associated factors were measured at a single time point. Recall bias may occur when collecting the associated factors based on the questionnaire. A longitudinal study is needed to investigate the long-term effects of implementing a kindergarten-based outreach program with SDF therapy on children’s DFA in the future. In addition, this research reflected the children’s DFA status after the dental examination and SDF therapy. Children’s psychological and anxious responses to both oral examination and SDF treatment could be overlapped. To determine the dental fear specifically due to SDF application, future study is required to investigate the relationship between dental fear of children who have dental examinations only and children who have dental examinations and receive SDF treatment.

This cross-sectional study showed that most children reported no or low DFA after receiving SDF therapy in the school-based outreach service. SDF therapy is a child-friendly caries treatment and could be one of the caries management alternatives reduce the burden of ECC. Notably, this study provided the information regarding children’s DFA after SDF therapy based on the kindergarten setting in Hong Kong; thus, the results may not be generalizable to other children in other geographic areas, other age groups or children receiving SDF in a clinical setting or undergoing some other complex treatment procedures.

## 5. Conclusions

Based on the child-reported DFA assessment, preschool children with ECC generally exhibited no or low DFA after SDF therapy in a school-based setting, regardless of their demographic background, dental visit and their DFA before SDF therapy.

## Figures and Tables

**Figure 1 ijerph-20-04467-f001:**
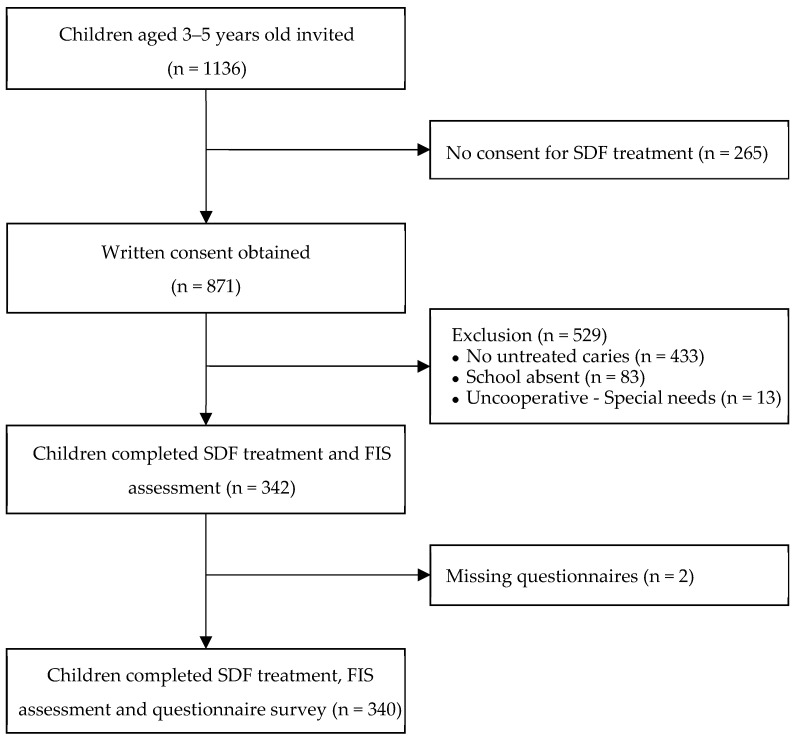
Children recruitment and participation in the study.

**Figure 2 ijerph-20-04467-f002:**
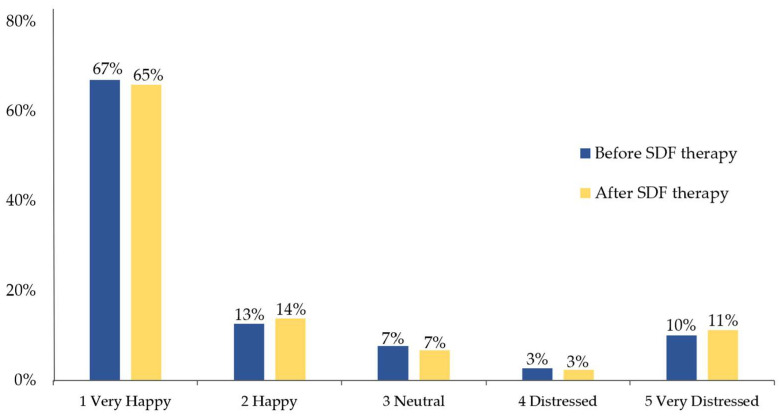
DFA of the study children before and after SDF therapy (n = 340).

**Table 1 ijerph-20-04467-t001:** Facial Image Scale (FIS).

Anxiety Level	5(Very Distressed)	4(Distressed)	3(Neutral)	2(Happy)	1(Very Happy)
Faces	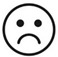	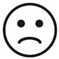	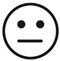	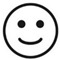	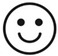

**Table 2 ijerph-20-04467-t002:** Characteristics of the study children and their parents (n = 340).

Characteristics	Frequency	Percentage
Child’s sex		
Male	187	55.0%
Female	153	45.0%
Child’s age		
3 years old	74	21.7%
4 years old	106	31.2%
5 years old	160	47.1%
Child’s birthplace		
Hong Kong	292	85.9%
Others	48	14.1%
Child’s dental visit		
Yes	71	20.9%
No	269	79.1%
Caries experience		
dmft ≤ 4	201	59.1%
dmft > 4	139	40.9%
DFA before SDF therapy		
No or low DFA	297	87.4%
High DFA	43	12.6%
Parents’ dental visit		
Yes	251	73.8%
No	89	26.2%
Mother’s education level		
Junior or senior education	67	19.7%
Tertiary education	273	80.3%
Father’s education level		
Junior or senior education	75	22.1%
Tertiary education	265	77.9%
Family’s monthly income (HKD)		
≤15,000	116	34.1%
>15,000	224	65.9%

**Table 3 ijerph-20-04467-t003:** Bivariate analysis between child’s Dental Fear & Anxiety (DFA) and potential factors (n = 340).

Potential Factors	No or Low DFA Number (%)	High DFA Number (%)	*p*-Value
Child’s sex			0.134
Male	157 (84%)	30 (16%)	
Female	137 (89%)	16 (11%)	
Child’s age			0.147
3 years old	64 (86%)	10 (14%)	
4 years old	97 (91%)	9 (9%)	
5 years old	133 (83%)	27 (17%)	
Child’s birthplace			0.496
Hong Kong	251 (86%)	41 (14%)	
Others	43 (90%)	5 (10%)	
Child’s dental visit			0.350
Yes	59 (83%)	12 (17%)	
No	235 (87%)	34 (13%)	
Caries experience			0.176
dmft ≤ 4	178 (89%)	23 (11%)	
dmft > 4	116 (84%)	23 (16%)	
DFA before SDF therapy *			0.801
No or low DFA	264 (89%)	33 (11%)	
High DFA	30 (70%)	13 (30%)	
Parents’ dental visit			0.988
Yes	217 (86%)	37 (14%)	
No	77 (87%)	12 (13%)	
Father’s education level			0.744
Junior or senior education	64 (85%)	11 (15%)	
Tertiary education	230 (87%)	35 (13%)	
Mother’s education level			0.242
Junior or senior education	55 (82%)	12 (18%)	
Tertiary education	239 (88%)	34 (12%)	
Family’s monthly income (HKD)			0.217
≤15,000	104 (90%)	12 (10%)	
>15,000	190 (85%)	34 (15%)	

* McNemar test. Others: Chi-square test.

## Data Availability

The datasets generated and/or analysed during the current study are available from the corresponding author upon reasonable request.

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
