# Peer review of "Dental Fear and Anxiety of Chinese Preschool Children in a School-Based Outreach Service Using Silver Diamine Fluoride for Caries Control: A Cross-Sectional Study"

_ijerph, 2023, doi:10.3390/ijerph20054467_

Round 1

Reviewer 1 Report

The non-invasive management of caries has received notable attention in recent years and in different settings, particularly in community dental care, care for non-cooperative children (due to immaturity) and those with special needs. Much of this interest is based on applying simple procedures that are better tolerated than other more interventional ones. Preschool patients, especially the youngest and sometimes with already complex pathologies (CEE caries), are strong candidates for them.

Under this premise, the reviewed article confirms what is already a clinical reality. The dental literature includes a large number of articles on the use of silver diamine fluoride (SDF) in children between 0-5 years of age. The rather predictable results of the work detract from interest.

It provides some originality in relation to the environment in which it is carried out (kindergartens) that offer a more familiar environment for preschoolers and consequently it is more "friendly" for the dental experience. Beyond the behavior of the healthy child, there are currently other areas that arouse the greatest interest in relation to this alternative (SDF) to the treatment of caries.

The article is basically well prepared, it presents some methodological doubts. Although the title relates dental fear and anxiety in preschoolers to the atraumatic caries control procedure (SDF), the methodology does not make it clear whether both psychological responses could overlap at the same time the first dental check-up and the application of the procedure (¿?). In this sense, it would be interesting to apply the questionnaire to children in whom dental exploration was carried out (many of them their first dental experience), but later dismissed for the study because they did not have cavities treatable with the procedure (SDF). The comparison between the explored-treated group versus the explored-untreated group would help to better discern the effect of the procedure on manifested fear and anxiety.

Regarding the variables studied in relation to the results obtained, there may be doubts regarding the relevance of some and the inclusion of others more directly related to the procedure and potentially related to some degree of rejection is missing, which could help to improve application protocols.

The discussion has been carried out with some methodologically very different studies, which detracts from the established comparison.

The results of the study, providing an overall positive experience, must be extrapolated with caution in future behavior in dental settings and especially in more complex procedures.

I believe that it can meet the minimum requirements for publication.

The non-invasive management of caries has received notable attention in recent years and in different settings, particularly in community dental care, care for non-cooperative children (due to immaturity) and those with special needs. Much of this interest is based on applying simple procedures that are better tolerated than other more interventional ones. Preschool patients, especially the youngest and sometimes with already complex pathologies (CEE caries), are strong candidates for them.

Under this premise, the reviewed article confirms what is already a clinical reality. The dental literature includes a large number of articles on the use of silver diamine fluoride (SDF) in children between 0-5 years of age. The rather predictable results of the work detract from interest.

It provides some originality in relation to the environment in which it is carried out (kindergartens) that offer a more familiar environment for preschoolers and consequently it is more "friendly" for the dental experience. Beyond the behavior of the healthy child, there are currently other areas that arouse the greatest interest in relation to this alternative (SDF) to the treatment of caries.

The article is basically well prepared, it presents some methodological doubts. Although the title relates dental fear and anxiety in preschoolers to the atraumatic caries control procedure (SDF), the methodology does not make it clear whether both psychological responses could overlap at the same time the first dental check-up and the application of the procedure (¿?). In this sense, it would be interesting to apply the questionnaire to children in whom dental exploration was carried out (many of them their first dental experience), but later dismissed for the study because they did not have cavities treatable with the procedure (SDF). The comparison between the explored-treated group versus the explored-untreated group would help to better discern the effect of the procedure on manifested fear and anxiety.

Regarding the variables studied in relation to the results obtained, there may be doubts regarding the relevance of some and the inclusion of others more directly related to the procedure and potentially related to some degree of rejection is missing, which could help to improve application protocols.

The discussion has been carried out with some methodologically very different studies, which detracts from the established comparison.

The results of the study, providing an overall positive experience, must be extrapolated with caution in future behavior in dental settings and especially in more complex procedures.

I believe that it can meet the minimum requirements for publication.

Author Response

Point 1: Although the title relates dental fear and anxiety in preschoolers to the atraumatic caries control procedure (SDF), the methodology does not make it clear whether both psychological responses could overlap at the same time the first dental check-up and the application of the procedure. In this sense, it would be interesting to apply the questionnaire to children in whom dental exploration was carried out (many of them their first dental experience), but later dismissed for the study because they did not have cavities treatable with the procedure (SDF). The comparison between the explored-treated group versus the explored-untreated group would help to better discern the effect of the procedure on manifested fear and anxiety.

Response 1: Agee. The limitations of the study design are discussed in 289–294.

Point 2: The discussion has been carried out with some methodologically very different studies, which detracts from the established comparison. 

Response 2: We appreciate the experts bringing these issues to our attention. This is the first research to explore children’s self-rated dental fear and anxiety among the children joining the dental outreach service using SDF for caries control. The number of articles on dental fear among preschool children receiving SDF is limited. Therefore, we needed to cite some related research with different methodologies to compare with our findings.

Point 3: The results of the study, providing an overall positive experience, must be extrapolated with caution in future behavior in dental settings and especially in more complex procedures. 

Response 3: Agree. Your suggestion is added to Lines 299–302. The generalisation of these findings to dental clinic settings should be taken with caution.

Besides revising the manuscript following the reviewer’s comments, please note that we were asked to add another 500 words to fulfill the journal’s requirement (to have at least 4000 words). The added texts were also highlighted in yellow.

Reviewer 2 Report

In general, the article proposes a very interesting topic with a potential for oral public health that has yet to be developed, namely the use of SDF for the treatment of caries in children. In principle, secondary aspects such as the control of fear and anxiety towards dental treatment should also be considered, and I congratulate the authors on their choice of subject matter for the article.

On a purely formal level, I would point out that the following remarks should be made:

TITLE

It should include the type of study that has been carried out.

It should also include the geographical area where the study was carried out, as fear and anxiety factors may be influenced by this variable.

INTRODUCTION

It points out that the SDF is a safe method, but it also has drawbacks for the treatment of caries that should be pointed out. It is not a 100% safe method and has only been cleared by the FDA for the treatment of hypersensitivity, not for the treatment of caries itself. Information and literature should be sought.

It should also introduce the disadvantages (not only aesthetic) of the use of SDF.

The objectives of the study should be much more clearly stated at the end.

MATERIAL AND METHOD

Dmft The term is misused according to the WHO. For primary dentition, the index to be used is dmft. If dmft is eventually used, it should be stated why.

For the use of this index, the caries criteria of the WHO oral health surveys publication should be used and the kappa index of the examiner(s) should be included.

Everything else is correct in the study.

Author Response

Point 1: On a purely formal level, I would point out that the following remarks should be made: 

TITLE 

It should include the type of study that has been carried out. 

It should also include the geographical area where the study was carried out, as fear and anxiety factors may be influenced by this variable. 

Response 1: Agree, the title was revised accordingly in Lines 2–4.

Point 2: It points out that the SDF is a safe method, but it also has drawbacks for the treatment of caries that should be pointed out. It is not a 100% safe method and has only been cleared by the FDA for the treatment of hypersensitivity, not for the treatment of caries itself. Information and literature should be sought. 

Response 2: Agree. The word "safe" is deleted, and the drawbacks and concerns of SDF are added in Lines 53–55 and 58–63.

Point 3: It should also introduce the disadvantages (not only aesthetic) of the use of SDF. 

The objectives of the study should be much more clearly stated at the end. 

Response 3: Done, the disadvantages are added in Lines 59–62 and 84–85, and the objective of the study is stated at the end in Lines 92–93.

Point 4: Dmft The term is misused according to the WHO. For primary dentition, the index to be used is dmft. If dmft is eventually used, it should be stated why. 

For the use of this index, the caries criteria of the WHO oral health surveys publication should be used and the kappa index of the examiner(s) should be included. 

Response 4: The clarification about dmft index with reference is added in Line 129-130.

The Kappa statistics is explained in the method part (Lines 141–143), and the results of Kappa are shown in Line 186.

Besides revising the manuscript following the reviewer’s comments, please note that we were asked to add another 500 words to fulfill the journal’s requirement (to have at least 4000 words). The added texts were also highlighted in yellow..
